# Characteristic changes in EEG spectral powers of patients with opioid-use disorder as compared with those with methamphetamine- and alcohol-use disorders

Christopher Minnerly[1,2], Ibrahim M. Shokry[1,3], William To[1], John J. Callanan[3], Rui Tao [1] *

1 Charles E. Schmidt College of Medicine, Florida Atlantic University, Boca Raton, Florida, United States of America, 2 FHE Health, Deerfield Beach, Florida, United States of America, 3 Ross University School of Veterinary Medicine, Basseterre, St. Kitts, West Indies

* rtao@health.fau.edu

**Data Availability Statement:** All relevant data are within the paper and its Supporting Information files.

## Abstract

Electroencephalography (EEG) likely reflects activity of cortical neurocircuits, making it an insightful estimation for mental health in patients with substance use disorder (SUD). EEG signals are recorded as sinusoidal waves, containing spectral amplitudes across several frequency bands with high spatio-temporal resolution. Prior work on EEG signal analysis has been made mainly at individual electrodes. These signals can be evaluated from advanced aspects, including sub-regional and hemispheric analyses. Due to limitation of computational techniques, few studies in earlier work could conduct data analyses from these aspects. Therefore, EEG in patients with SUD is not fully understood. In the present retrospective study, spectral powers from a data house containing opioid (OUD), methamphetamine/stimulants (MUD), and alcohol use disorder (AUD) were extracted, and then converted into five distinct topographic data (i.e., electrode-based, cortical subregion-based, left-right hemispheric, anterior-posterior based, and total cortex-based analyses). We found that data conversion and reorganization in the topographic way had an impact on EEG spectral powers in patients with OUD significantly different from those with MUD or AUD. Differential changes were observed from multiple perspectives, including individual electrodes, subregions, hemispheres, anterior-posterior cortices, and across the cortex as a whole. Understanding the differential changes in EEG signals may be useful for future work with machine learning and artificial intelligence (AI), not only for diagnostic but also for prognostic purposes in patients with SUD.

## Introduction

EEG was discovered in the 1920's, and explored for biomedical purposes since the 1930's [1,2]. Signals are primarily derived from cortical pyramidal neurons that generate postsynaptic

**Funding:** The authors received no specific funding for this work.

**Competing interests:** NO authors have competing interest

potentials propagated towards the apical dendrites perpendicular to the cortical surface. Graphic waves vary irregularly, reflecting the net change between inhibitory and excitatory postsynaptic potentials in a temporal- and spatial-dependent manner. Thus, unlike magnetic resonance imaging (MRI) or positron emission tomography (PET), raw data are hardly interpretable, but require decomposition and then further reorganized into graphic images [3]. Current computational methods make it possible for signals to be easily reprocessed and transformed into interpretable spectral images into at least one of three distinct methods. The most common method is to analyze constituents of spectra (i.e., frequency bands) including delta/δ, 0.1–4.0 Hz; theta/θ, 4.0–8.0 Hz; alpha/α, 8.0–12.0 Hz; beta/β, 12.0–25 Hz; gamma/γ, >25 Hz. Changes in frequency bands are commonly used in sleep and arousal investigation. Compared to healthy adults, patients with an arousal disorder had an excessive amount of slow-wave sleep (SWS; mainly delta/δ waves) interruption [4]. The next method is the event-related potential (ERP) by determining the signal-to-noise ratio of the EEG signals at a given time associated with a specific stimulus, which has become a popular tool in the study of sensory, cognitive, or motor events. For example, positive potentials at 300 msec (P300) are currently used for a biomarker for schizophrenia examined in clinical settings [5–8]. Thirdly, EEG signals are transformed and quantified into a color-specific topographic map, which is associated with respective cortical activity [9,10]. In epilepsy, topographic images provide a guide to remove epileptogenic zones during brain surgery [11,12].

EEG as a powerful tool used to study opioid- (OUD), methamphetamine- (MUD), and alcohol-use disorders (AUD) has been ventured over the past few decades. Most efforts were made to identify frequency bands in relationship with EEG potentials in the closed-eye (i.e., resting) state. Opioid abuse can cause a loss of GABAergic inhibitory control over postsynaptic excitatory potentials, including cortical pyramidal neurons ([13]; also reviewed by Baldo et al, 2016 [14]), resulting in an alteration of electrical synchronization between cortical neurons. By analysis of delta/δ, theta/θ, alpha/α, beta/β, and gamma/γ waves, it was found that all of the five spectral powers was elevated with almost equipotency in the frontal, central, temporal, parietal, and occipital subregions of patients with OUD [15]. However, others demonstrated that it was only certain spectra, but not all, that were elevated in the cortical subregions [16–19]. The selective effects are also reported in MUD and AUD. Methamphetamine (METH) exposure for a long period time may cause a reduction in dopamine transporters in the brain [20]. Newton et al (2003) showed that the delta/δ and theta/θ bands, but not others, were elevated almost globally in the cortical subregions [21]. The findings were partly supported by Khajehpour et al (2019), showing that delta/δ and gamma/γ powers were slightly, yet significantly, increased in a topographic analysis [22]. Alcohol is believed to be inhibitory, mimicking GABA's effect on postsynaptic $GABA_A$ receptors [23]. The gamma/γ powers, but not other frequency bands, were elevated across the cortex of patients with AUD [24]. However, Ko and Park showed that there was a reduction in alpha/α power while an increase in gamma/γ powers [25]. Interestingly, by analyzing EEG obtained from 191 male alcoholic patients, Coutin-Churchman et al. revealed that the most frequent reduction took place in the delta/δ and theta/θ bands [26]. Nevertheless, although EEG has been used as a tool to estimate mental health, there has been no consensus on spectral powers altered in patients with SUD (i.e., OUD, MUD, or AUD).

To this end, it appears that EEG signals were rarely examined from a topographic view over EEG spatial sizes in terms of areas covered by electrodes, such as the size at individual electrodes, regional, hemispherical, or frontal-posterior till the brain as a whole. We proposed to test that EEG spatial sizes are attributed to credential of data analysis. With this goal in mind, EEG signals obtained from 19 electrodes and 6 cortical areas were first broken down into 5 spectra (i.e., delta/δ, theta/θ, alpha/α, beta/β, and gamma/γ), and then re-arranged, combined,

and remapped topographically. EEG signal decomposing and rearranging would reveal the topographic difference between drugs. In this work, we sought to characterize EEG signals in patients with OUD in contrast to those obtained from MUD or AUD. An advantage of the comparative study was that EEG was collected at the same rehabilitation facility and thus, data treatment was standardized for each group.

## Materials and methods

### Patients

Data were obtained from an electronic medical data house at a substance abuse treatment facility (FHE Health, Deerfield Beach, FL, USA), which had gathered ~1000 cases of information about patients' drug use history, DMS-5 diagnosis, and drug intoxication treatment. In addition, there were 20 cases obtained from healthy subjects with no substance abuse history. EEG data prior to treatment were tracked electronically, along with information about detox-related symptoms. Searches with opioid-related keywords (i.e., morphine, heroin, fentanyl, methadone or oxycodone) found 350 patients who had records of opioid use history. Approximately 450 patients had records of alcohol use history. Methamphetamine-related keywords (i.e., crystal meth, meth, ice) yielded approximately 100 records of METH use history, while the remaining cases were a mix of substance use disorders. To this end, thirteen men and seven women identified as OUD were compared with 20 sex- and age-matched healthy controls (Table 1); fifteen patients identified as MUD; and twenty-three as AUD were compared to those with OUD. All data were fully anonymized before access (see the S1–S5 Files), and the requirement for informed consent for retrospective data analysis was waived by the institutional review board (IRB) from Florida Atlantic University (Boca Raton, FL, USA) and Ross University School of Veterinary Medicine (St. Kitts, West Indies).

### EEG data acquisition

EEG recordings were performed between 12:00 PM—4:00 PM. Following instrumental calibration, the subject (patient or healthy control) was seated in a comfortable chair in a dimmed recording room and the EEG procedures were orally instructed. A cap with 19 electrodes (Electro-Cap International, Eaton, OH, USA) was placed on the scalp (Fig 1A). To reduce muscle artifacts in the EEG signal, the participant was instructed to assume a comfortable position. Eyes closed, and subject seated calmly without any activity or movement. Signals were collected with the band-pass filter of 1–100 Hz at a rate of 256 Hz, and amplified with Neurofield's Q20 amplifier (NeuroField Inc., Bishop, CA, USA; Fig 1B) using NeuroGuide software (Applied Neuroscience Inc., Tampa, FL, USA). Each subject underwent 10 minutes of EEG recording with eyes closed.

**Table 1. Health profile of subjects used in the studies.**

|  | CTL (N = 20) | OUD (N = 20) | MUD (N = 15) | AUD (N = 23) | P (*vs*. CTL) |
|---|---|---|---|---|---|
| Age (years) | 33 (±12) | 34 (±12) | 29 (±8) | 38 (±10) | >0.05 |
| Sex (M/F) | 13/7 | 13/7 | 11/4 | 17/6 | n/a. |
| Duration of substances used (years) | 0 | 7 (±5) | 5 (±3) | 9 (±7) | <0.05 |
| Substances used | no | Morphine; heroin, oxycodone. | METH | alcohol | n/a. |

n/a.; not applicable.

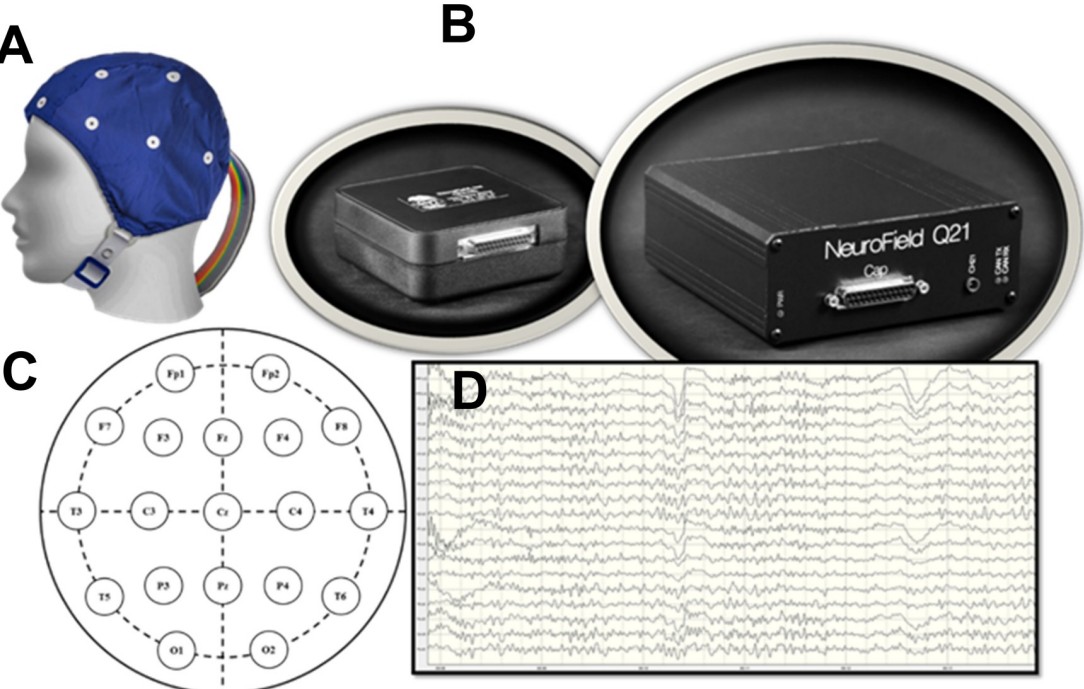

**Fig 1. EEG data acquisition.** (**A**) A 19-channel EEG cap (from Electro-Cap International, Inc. Eaton, OH, USA) used for collecting data. (**B**) QCheck electrode impedance monitor and Q21 amplifier (Neurofield, Inc., Bishop, CA, USA). (**C**) A diagram of the International 10–20 System to elaborate electrode placement across the scalp. (**D**) An example of a digitized EEG recording using Neuroguide software (Applied Neuroscience, Inc. Largo, FL, USA).

### EEG data analysis and rationale for five distinct approaches

Only one time of data prior to rehab treatment was approved by the IRAs to be used in the present study. EEG data were downloaded from the database as described previously [18]. Briefly, raw data was edited using the editing tool within the NeuroGuide software to remove physical artifacts (including eye movement, jaw movement, and gross movement) and was then visually inspected. A 60-second epoch of quality data was gathered after removal of the aforementioned artifacts. Epoch selection was governed by reliability measures of the data within the NeuroGuide program. Test-retest values of 0.90 or greater are considered highly reliable and valid according to literature [27]. Each epoch was subjected to EEG spectral power analysis, using a fast Fourier transform (FFT), and then extracted to Microsoft Excel for further data calculation. Powers of delta/δ (1–4 Hz), theta/θ (4–8 Hz), alpha/α (8–12 Hz), beta/β (12-25Hz), and gamma/γ (25–50 Hz) oscillations were individually sorted according to electrodes and averaged (mean ±SEM).

EEG signals, consisting of 5 spectral powers and 19 electrodes, were characterized in five distinct ways (Table 2). First, spectral powers at individual electrodes between healthy controls (CTL) and SUD were directly used for data comparison and analysis. This approach has been widely employed by many laboratories previously [for instance, [17,19]], and thus defined as Approach 1. The advantage of using Approach 1 was that no computation was required in the data analysis. However, since limited spatial sizes, one would often find that changes in EEG signals were significantly altered at some electrodes but not others. Given this, it could be difficult to draw conclusions of what happened in EEG signaling. To solve this problem, new approaches of data analysis developed from four additional approaches. Specifically at Approach 2, spectral power data from several electrodes were grouped into 1 (prefrontal; Fp1

**Table 2. Comparison of Approach 1–5 used in the present studies.**

| Approach | Main Features | Advantage | Disadvantage |
|---|---|---|---|
| 1 | Electrode-based analysis | Less computation needed Commonly used; Reference literature available | A huge amount of end data; Hard to find difference between CTL and SUD |
| 2 | Cortex-based analysis | EEG signals associated with specific cortices; Easy to find difference between CTL and SUD | Lack of details in EEG signals; Amount of end data is still huge; Computational analysis needed |
| 3 | Hemisphere-based analysis | Only two sets of end data; Easy to find difference between CTL and SUD | Part of EEG signals excluded from analysis; Comprehensive computation needed |
| 4 | Anterior-posterior analysis | Only two sets of end data; Easy to find difference between CTL and SUD | Part of EEG signal excluded from analysis; Comprehensive computation needed |
| 5 | Total cortex-based analysis | A single set of end data; Easy to find difference between CTL and SUD | Lack of detailed information; Likely misdiagnosis |

and Fp2), 2 (frontal; F3, F4, F7, F8, and Fz), (central; C3, C4, Cz, T3, and T4), 4 (temporal; T5 and T6), 5 (parietal; P3, P4, and Pz), and 6 (occipital; O1 and O2). Next, EEG signals were viewed from the hemispheric level designated as Approach 3. Spectral data of Fp1, F3, F7, C3, T3, T5, P3, and O1 were grouped and expressed as mean ± SEM into data 1, and Fp2, F4, F8, C4, T4, T6, P4, and O2 into data 2 as the left and right, respectively, hemispheric subregions. Note that data from the central subregions (Fz, Cz, and Pz) were excluded from the analysis. Next, EEG signals were viewed from an anterior-posterior aspect designated as Approach 4. EEG signals obtained from Fp1, Fp2, F3, F4, Fz, F7, and F8 were grouped as mean ± SEM representing anterior EEG activity, while O1, O2, P3, P4, Pz, T5, and T6 grouped as the posterior EEG activity. Note T3, T4, C3, C4, and Cz were excluded from the data analysis. Lastly, spectral data was viewed as a whole, across the cortex designated as Approach 5. All of 19 electrodes was grouped as mean ± SEM.

## Statistical analysis

Data are expressed as mean ± SEM, and evaluated with repeated measures ANOVA between CTL and SUD (OUD, MUD, and AUD) followed by *post-hoc* Fisher's PLSD test using Stat-View software 5.0 (SAS Institute Inc., Cary, NC, USA). If appropriate, unpaired Student *t*-test was also utilized to determine statistical difference. Significance was set at $P < 0.05$.

## Results

### Characterization of EEG spectral powers at cortices of the healthy brains

Spectral power data obtained from 20 healthy controls with 19-channel caps are grouped into 5 bands (delta/$\delta$, 1–4 Hz; theta/$\theta$, 4–8 Hz; alpha/$\alpha$, 8–12 Hz; beta/$\beta$, 12–30 Hz; and gamma/$\gamma$, 30–50 Hz), and further classified into 6 subgroups: the prefrontal (Fp; Fp1 and Fp2), frontal (F; F3, F4, F7, F8, and Fz), central (C; C3, C4, Cz, T3, and T4), temporal (T; T5, T6), parietal (P; P3, P4, and Pz), and occipital (O; O1 and O2). Statistical analysis reveals that amplitudes of spectral powers ($\mu V^2$) of those 5 bands are significantly different in 6 cortical subregions [delta/$\delta$, $F_{(5,374)} = 8.25$, $P < 0.0001$; theta/$\theta$, $F_{(5,374)} = 3.817$, $P = 0.0022$; alpha/$\alpha$, $F_{(5,374)} = 9.185$, $P < 0.0001$; beta/$\beta$, $F_{(5,374)} = 9.185$, $P < 0.0001$; gamma/$\gamma$, $F_{(5,374)} = 2.969$, $P = 0.0121$). As shown in Fig 2, the y-axis displays spectral powers plotted against 6 cortical subregions displayed in x-axis. Except for the delta/$\delta$ band, the greatest spectral powers of theta/$\theta$, alpha/$\alpha$, beta/$\beta$, and gamma/$\gamma$ were found in the occipital (O) cortex. In contrast, the greatest delta/$\delta$ powers (Fig 2A) were in the prefrontal area, followed by the frontal, central, parietal, occipital, and temporal subregions. Interestingly, there exhibited a characteristic distribution of spectral power levels. As shown in the right panel of Fig 1A, the delta/$\delta$ power levels went from greatest

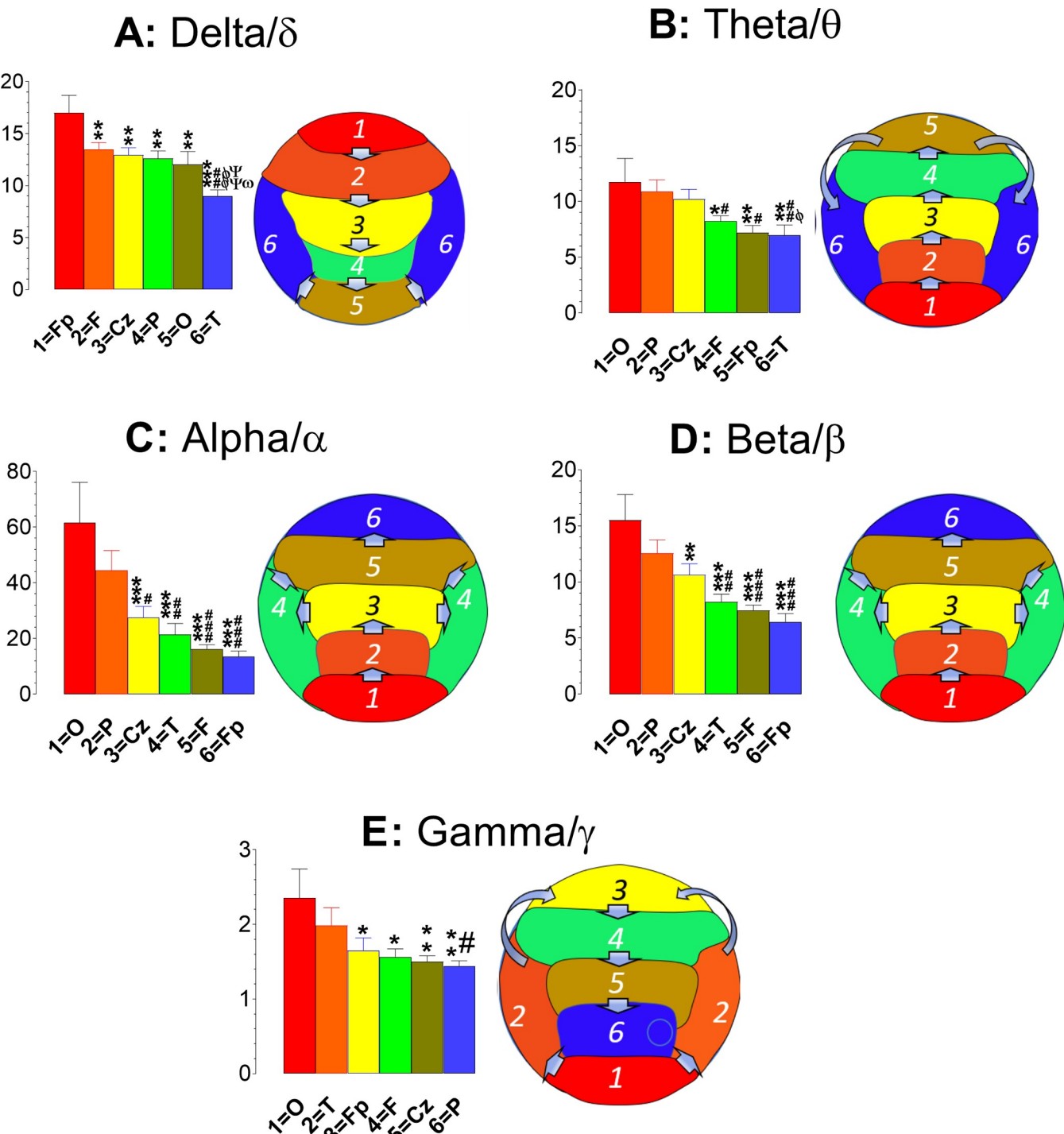

**Fig 2. Topographic analysis of the EEG bands at cortical subregions of the healthy brains (N = 20).** The y-axis indicates spectral powers ($\mu V^2$) plotted against 6 cortical subregions displaying in x-axis. Data are expressed as the rank orders from the highest to lowest powers in the subregions. Except for delta/$\delta$ waves (**A**), the highest amplitude powers were found in the occipital subregion with characteristic rank orders. Specifically, the theta/$\theta$ powers (**B**) were found in a rank order of O →P →Cz → F →Fp →T). * P<0.05 and ** P<0.01 *vs.* 1 = O; #P<0.05 and ##P<0.01 *vs.* 2 = P; $\phi$P<0.05 vs. 3 = Cz. The rank orders for alpha/$\alpha$ (**C**) and beta/$\beta$ (**D**) were identical, displaying O →P →Cz →T →F →Fp. **P<0.01 and ***P<0.001 *vs.* 1 = O; #P<0.05, ##P<0.01 and ###P<0.001 *vs.* 2 = P. Interestingly, the gamma/$\gamma$ powers were the highest at the occipital subregion and made a turn to the temporal lobe and then the prefrontal subregion, and finally ended at the lowest power in the parietal subregion. **P<0.01 vs. 1 = O; #P<0.05 vs. 2 = T. In contrast, the highest amplitude powers for delta/$\delta$ waves were in the frontal subregions followed by rear subregions, and then the temporal lobes (Fp →F →Cz →P →O →T). **P<0.01 and ***P<0.001 *vs.* 1 = O; ##P<0.01 *vs.* 2 = F; $\phi\phi$P<0.01 *vs.* 3 = Cz; $\Psi\Psi$P< *vs.* 4 = P; $\omega$P<0.05 *vs.* 5 = O.

to least in the anterior to posterior subregions and then to the lateral lobes (Fp →F →Cz →P →O →T). In contrast, theta/θ powers took nearly the opposite direction, from the posterior to anterior subregions and then to the lateral lobes (Fig 2B; O →P →Cz → F →Fp →T). Alpha/α and beta/β powers had an identical direction of ranking orders, showing the posterior to central and lateral subregions, and finally to the anterior lobes (Fig 2C and 2D; O →P →C →T →F →Fp). The direction utilized by gamma/γ was relatively complicated but still followed a pattern, showing the ranking orders from the occipital cortex to the lateral and then to the prefrontal cortex, from where direction changed to the central subregions (Fig 2D; O →T →Fp →F →C →P).

## Approach 1: Electrode-based analysis

Fig 3A displays a representative delta/δ wave at the F3 electrode obtained from a healthy control (CTL) compared with individuals with OUD, MUD, or AUD. Compared to CTL, delta/δ amplitudes were increased in patients with OUD or MUD, but not AUD. On the contrary, it was reduced in the AUD case. Next, the EEG waves were transformed into amplitude powers expressed as $\mu V^2$, as shown in the left panel of Fig 3B. The difference in delta/δ amplitude powers was statistically significant [$F_{(3, 74)} = 6.07$, $P = 0.0009$]. However, *post-hoc* analysis indicates that only OUD, but not MUD or AUD, reached statistical significance difference from the CTL. Analysis of absolute values could be simple but potentially create data bias due to individual differences. To minimize such possibility, data were normalized into % CTL. As shown in the right panel of Fig 3B, changes at the F3 electrode were approximately 50%, 30%, and -30% relative to the CTL, in OUD, MUD, and AUD, respectively. The normalized data appear to have the same response pattern as the absolute values. Moreover, statistical analysis reached the same conclusion as using the absolute values. For a sake of comparison between our results and other labs, data described below are expressed mainly as absolute values.

Next, we analyzed all 19 individual electrodes. Compared with the CTL, there were at least 10% increases in delta/δ powers of patients with OUD, but not MUD or AUD. Specifically for OUD, 14 electrodes (73%) displayed at least 50% higher power than the CTL. However, only 7 electrodes (i.e., F3, F4, C3, C4, T4, P4, and Pz) reached statistical significance (P <0.05; ANOVA; Fig 3C–3G). Although the rest of electrodes had no significant changes, their delta/δ powers still followed the same pattern as indicated with the dash lines on the graphs.

Finally, effects of SUD on theta/θ (S6 File), alpha/α (S7 File), beta/β (S8 File), and gamma/γ waves (S9 File) were compared with the CTL. Although there was a tendency, no statistically significant difference from the CTL was found any individual electrode (P >0.05; ANOVA).

## Approach 2: Cortex-based analysis

Spectral power data are grouped into 1 (prefrontal; Fp1 and Fp2), 2 (frontal; F3, F4, F7, F8, and Fz), 3 (central; C3, C4, Cz, T3, and T4), 4 (temporal; T5 and T6), 5 (parietal; P3, P4, and Pz), and 6 (occipital; O1 and O2). Fig 4 displays spectral powers ($\mu V^2$) of those cortical subregions obtained from CTL compared with patients with OUD, MUD, or AUD. Compared to CTL, it appears that OUD and MUD had elevated spectral powers for delta/δ and theta/θ, while reduced in alpha/α powers. However, beta/β or gamma/γ powers could not be clearly determined with the analysis used in Approach 2. Statistical analysis reveals significant increases in delta/δ powers [$F_{(3, 74)} = 6.753$, $P = 0.0004$], had no effect on theta/θ [($F_{(3, 74)} = 2.224$, $P = 0.0924$); alpha/α ($F_{(3, 74)} = 1.605$, $P = 0.1955$); beta/β, ($F_{(3, 74)} = 0.732$, $P = 0.5359$); gamma/γ, ($F_{(3, 74)} = 0.732$, $P = 0.5359$)].

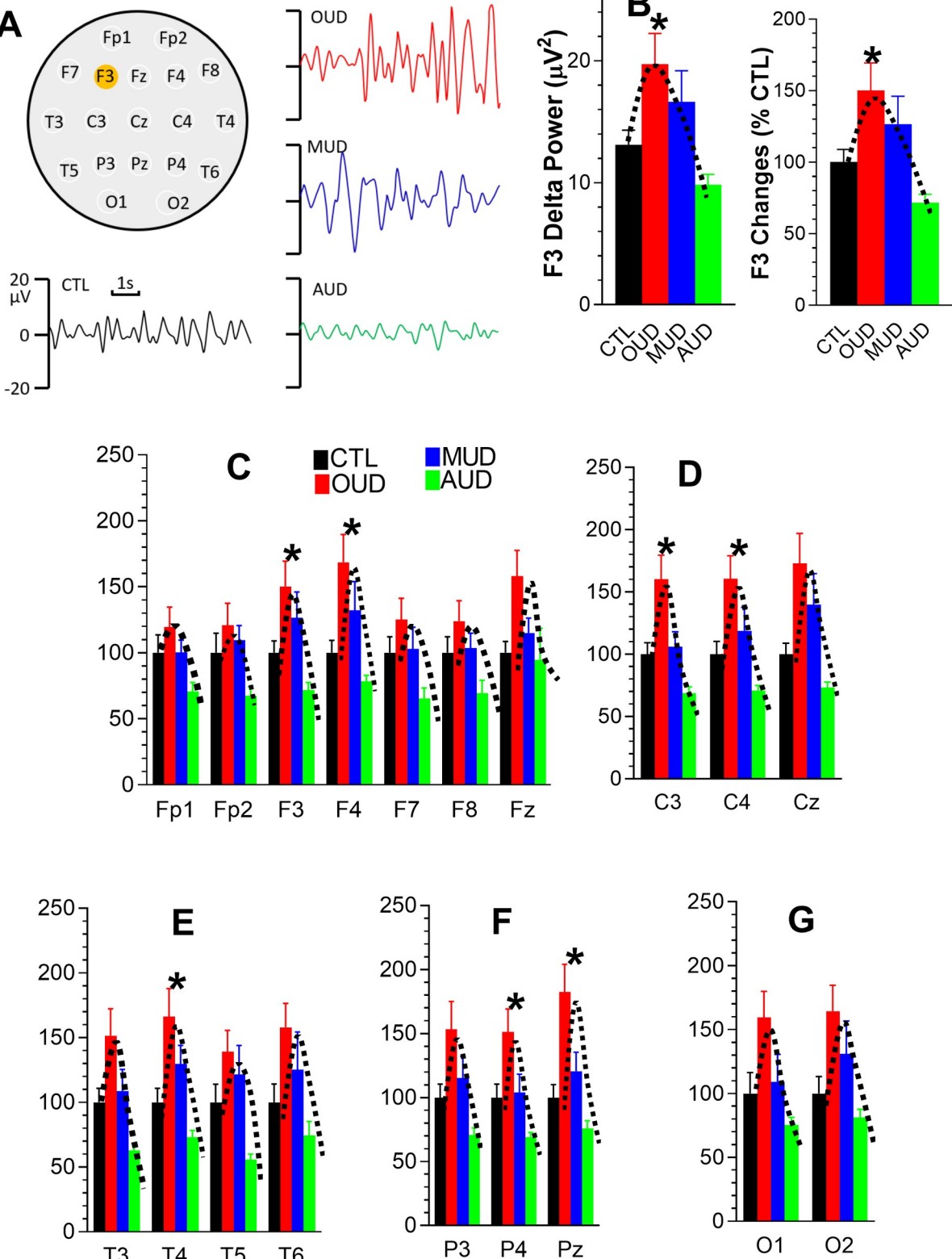

**Fig 3. Effects on delta/δ powers at 19 individual electrodes of patients with OUD, MUD or AUD. A**, Representative delta/δ waves in the F3 electrode from CTL, OUD, MUD and AUD. **B**, F3 delta/δ powers expressed as absolute values ($\mu V^2$; left panel) or 100% CTL level (right panel). **C**, Frontal delta/δ powers. **D**, Central delta/δ powers. **E**, Temporal delta/δ powers. **F**, Parietal delta/δ powers. **G**, Occipital delta/δ powers. *P <0.05 *vs*. CTL.

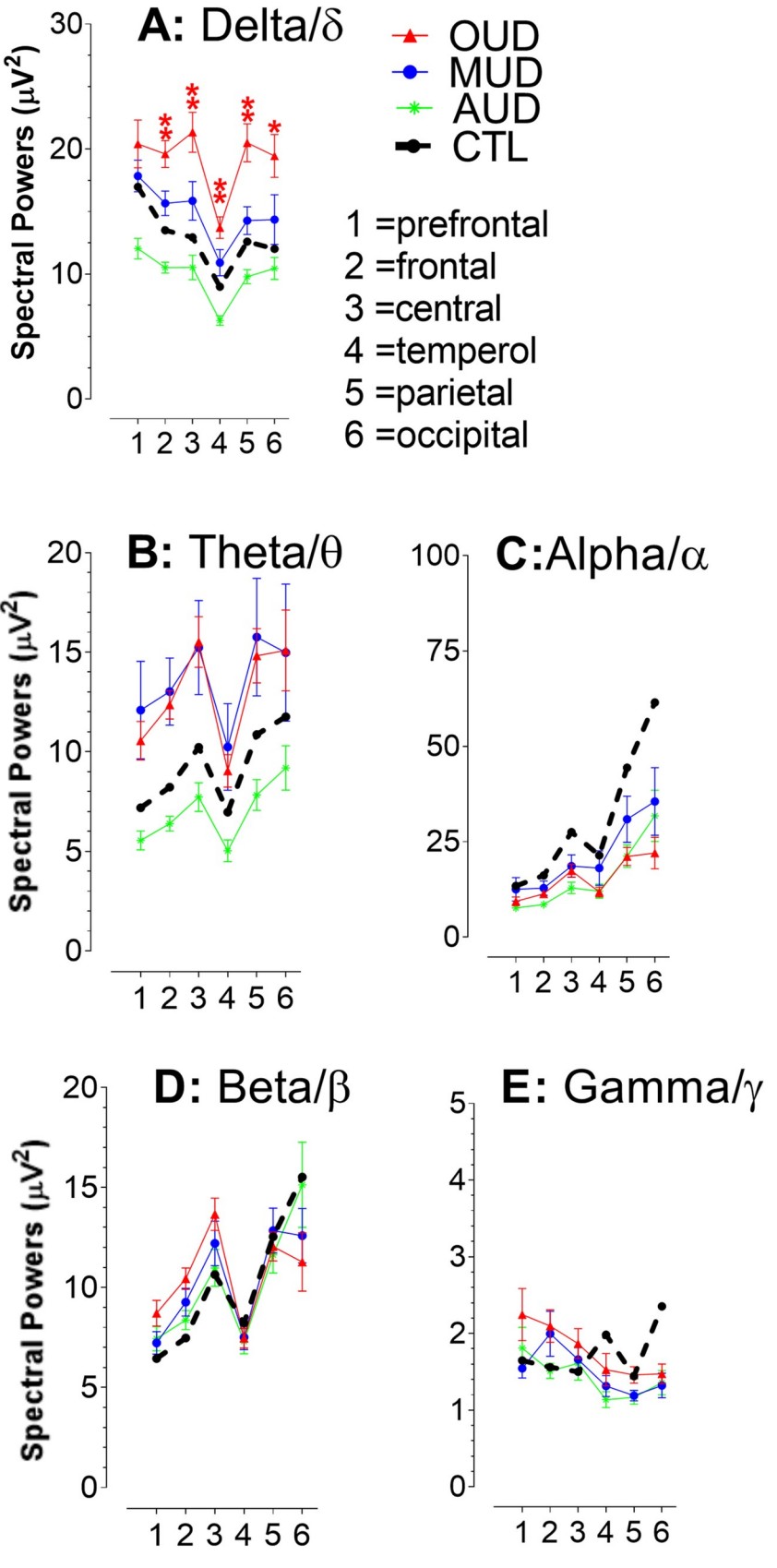

**Fig 4. Phenotypic changes of spectral powers in cortical subregions.** Numbers in x-axis indicate cortical subregions. 1, prefrontal; 2, frontal; 3, central; 4, temporal; 5, parietal; 6, occipital. In the case of OUD and MUD, delta/δ (**A**) and theta/θ powers (**B**) appeared to be elevated in all 6 cortical subregions as compared to CTL. In contrast, alpha/α powers (**C**) were lower than the CTL. There was no clear pattern for beta/β (**D**) or gamma/γ powers (**E**). Regarding AUD, spectral powers for delta/δ, theta/θ and alpha/α bands were lower than the CTL. No clear pattern for beta/β or gamma/γ bands was found. $^*P{<}0.05$, and $^{**}P{<}0.01$ *vs.* CTL, a post-hoc Fisher's PLSD test followed by ANOVA.

## Approach 3: Analysis of the left-right hemisphere axis and spectral powers

In this section, spectral data were grouped and expressed as mean ± SEM into 1 and 2, respectively, representing the left and right hemispheres. Note that data from the central subregions (Fz, Cz, and Pz) were excluded from the analysis. Fig 5 displays spectral powers obtained from healthy controls (CTL) compared with individuals with OUD, MUD, or AUD. Analysis was performed from two aspects. First, we found that spectral powers of two hemispheres were almost at the same level, parallel to the x-axis. This suggests that substance use disorders (OUD, MUD, or AUD) did not have a selective effect on hemispheres. We next determined effects of substance use on spectral powers by analysis of y-axis. Compared to CTL, OUD and MUD had an increased power of delta/δ (**A**) and theta/θ (**B**), but a decreased alpha/α power (**C**). In contrast, AUD showed a reduction in all three waves. However, only delta power reached statistical significance [left, $F_{(3,620)} = 36.748$, P $<0.0001$; right, $F_{(3,620)} = 36.694$, P $<0.0001$). Changes in beta/β (**D**) or gamma/γ powers (**E**) were not statistically significant.

## Approach 4: Analysis of the anterior-posterior axis and spectral powers

Data obtained from Fp1, Fp2, F3, F4, Fz, F7, and F8 were grouped as mean ± SEM representing for anterior signals, while O1, O2, P3, P4, Pz, T5, and T6 grouped as the posterior activity. Note T3, T4, C3, C4, and Cz were excluded from the data analysis (Fig 6A).

First, the x-axis (horizontal) levels were analyzed. In the CTL group, we found that, except for the delta/δ wave, the powers of theta/θ, alpha/α, beta/β, and gamma/γ were greater at posterior regions than those at the anterior regions. However, powers of delta/δ at the anterior regions were greater. We found that, except for the gamma/γ wave, the drug use disorders (OUD, MUD, or AUD) did not alter the relationship of the anterior-posterior axis. However, such relationship had been reversed in the gamma/γ wave (**E**), showing that the anterior powers were elevated while posterior powers were reduced. Next, we conducted statistical analysis on the y-axis. There were significant main effects on the delta/δ [**A**; 1 = anterior, $F_{(3,542)} = 26.001$, P$<0.0001$; 2 = posterior, $F_{(3,542)} = 36.308$, P$<0.0001$] and theta/θ waves [**B**; 1 = anterior, $F_{(3,542)} = 21.036$, P$<0.0001$; 2 = posterior, $F_{(3,542)} = 9.675$, P$<0.0001$]. Changes in alpha/α (**C**), beta/β (**D**), or gamma/γ (**E**) were not significant.

Since the anterior-posterior relationship in gamma/γ waves were reversely altered in patients with SUD (i.e., OUD, MUD, or AUD), it prompted us to determine whether the reversed effect was statistically significant at individual cortices. Thus, the gamma/γ data were decomposed, and then regrouped to 6 cortical subregions. As shown in Fig 7A, significant changes occurred in the frontal [$F_{(3,386)} = 2.694$, P = 0.0458), temporal ($F_{(3,308)} = 4.18$, P = 0.0064), and occipital ($F_{(3,152)} = 4.225$, P = 0.0067), but not prefrontal ($F_{(3,152)} = 1.382$, P = 0.2505) or central subregions ($F_{(3,230)} = 3.067$, P = 0.0285]. Topographic analysis (Fig 7B) revealed that the lowest gamma/γ power still remained at the parietal subregion. However, the highest gamma/γ power was drifted towards prefrontal (OUD and AUD) or frontal subregions (MUD), resulting in changes in the anterior-posterior relationship.

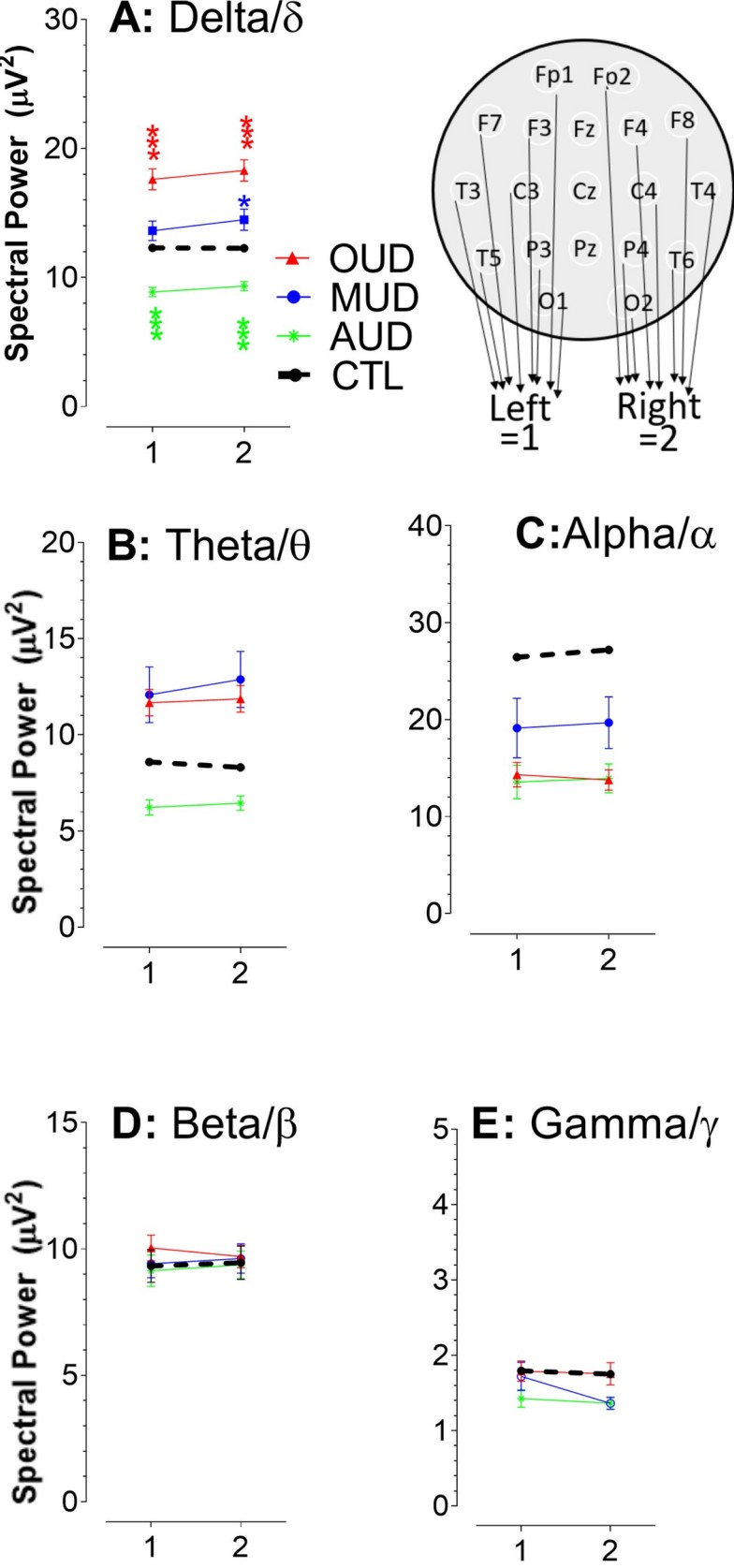

**Fig 5. Left hemispheric spectral powers compared with the right hemispheric subregions.** Numbers in x-axis denote the left and right hemispheres as 1 and 2, respectively. Data are expressed as mean ± SEM. Compared to CTL, OUD and MUD had an elevated power of delta/δ (**A**) and theta/θ (**B**) waves but a reduced alpha/α wave (**C**). In contrast, all three waves were reduced in AUD. No change was observed as the data expressed as hemispheric beta/β (**D**) or gamma/γ powers (**E**).

## Approach 5: Analysis of spectral powers across total cortex

Data obtained from 19 electrodes were grouped as mean ± SEM representing for spectral power across the whole cortex. ompared to CTL, spectral powers in patients with SUD (OUD, MUD or AUD) were significantly altered in delta/δ and theta/θ, partly alpha/α or gamma/γ (Fig 8). No effect was observed in the beta/β.

## Discussion

The present study revealed that EEG signals can be decomposed into many elements and regrouped into multiple datasets, showing characteristic patterns in patients with OUD compared to those with MUD, AUD, or healthy controls (CTL). It appears that data regrouping and reanalyzing had no effect on the EEG patterns, but markedly increased EEG credentials. To obtain an unbiased conclusion, we therefore suggest that the EEG signals are best viewed from 5 distinct perspectives, including from an individual electrode aspect, a cortical subregion level, a left-right hemispheric axis, an anterior-posterior axis, and the cortex as a whole.

EEG signals were analyzed with 5 approaches (Table 2). Approach 1 (electrode-based) analysis has been widely used to determine EEG activity [16,17,28] because of simplicity without additional computation. Despite tendency, however, changes in EEG signals at individual electrodes often fail to reach statistical significance (see details in Fig 3, and also S1–S4 Files). One explanation for such failure could be that the retrospective analysis was not designed as a one-to-one matched case control study. It was likely to be that the statistical power would be increased if the subgroups were matched. To test such possibility, subjects were reduced to 15 cases/subgroup for the age- and sex-matched design. Delta/δ powers presented in Fig 3B were re-examined with the ANOVA, revealing significant difference between 4 groups, $F_{(3,56)} = 3.203$, $P = 0.03$. However, the *post-hoc* analysis revealed that changes in OUD, MUD or AUD were not different from CTL, suggesting the one-to-one matched design could not fully explain the failure at statistical analysis.

Alternatively, a single electrode covered only small brain areas in terms of EEG spatial sizes and thus relatively small numbers of neurons affected. Sample sizes, which are a major obstacle in the human studies, were likely attributed to the failure in statistical evaluation. To this point, it appears that EEG spatial sizes or sample sizes are critical for revealing significance of EEG data acquired. It has been suggested that adjacent electrodes are functionally coherent although such relationship for two distant electrodes, particularly at different cortical subregions, does not exist [29,30]. Findings that EEG amplitudes of adjacent electrodes were at the same level [18] support the coherent hypothesis. This suggests that, despite different electrodes but physical adjacency, their spectral powers could be grouped to determine functional changes. Taken together, combination of adjacent electrodes, which would increase not only spatial sizes but also sample sizes, may help improve EEG credentials and statistical analysis. Taking advantage of this concept, individual electrodes were selectively combined and regrouped according to cortices and expressed as six subregions using Approach 2 (see details in Fig 4). As a result, it was clearly demonstrated that EEG signals, particularly on delta/δ, were synchronized in patients with OUD and MUD and desynchronized with AUD. However, the significant difference took place only in OUD patients. Changes in EEG signals became more

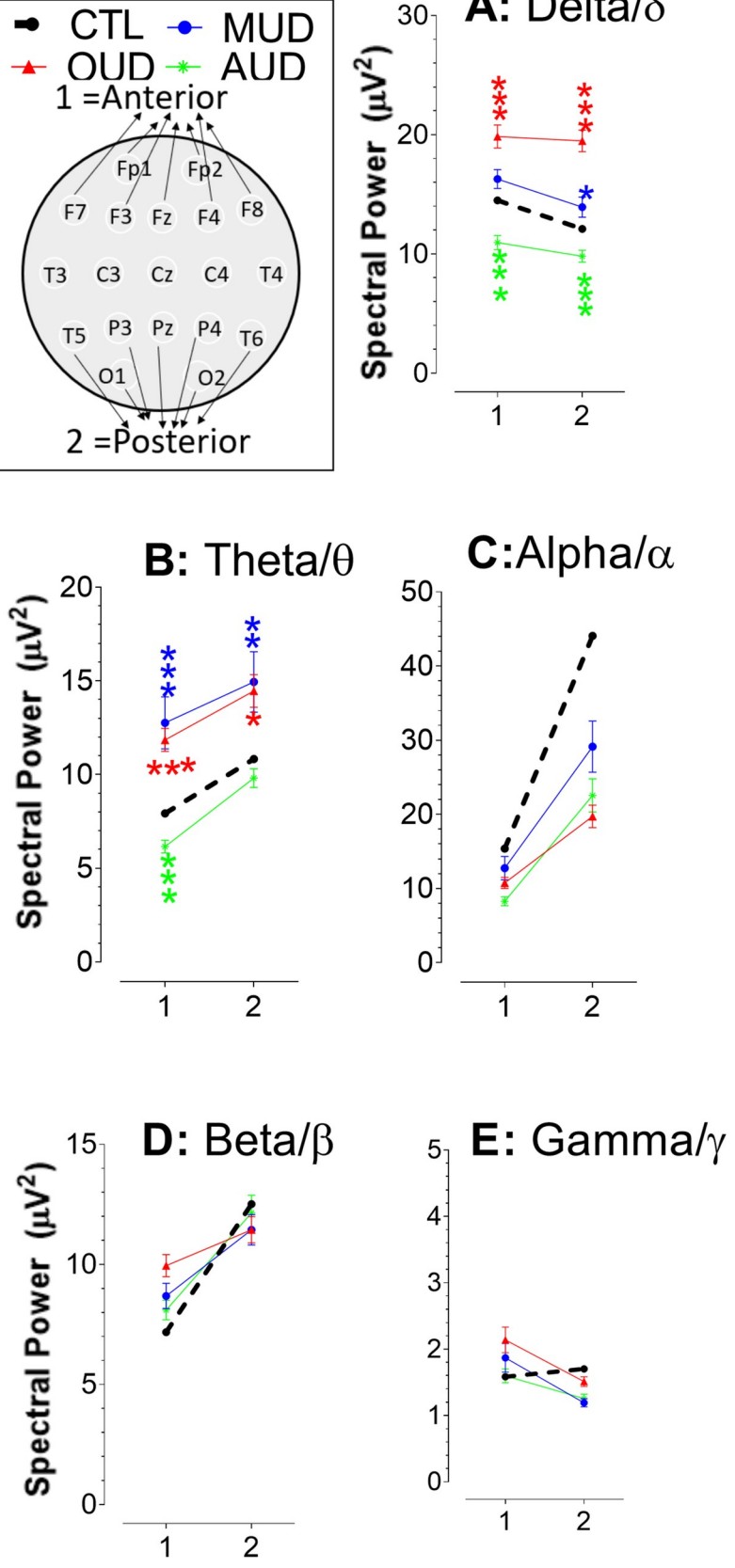

**Fig 6. Anterior spectral powers compared with the posterior subregions.** Numbers in x-axis denote the anterior and posterior powers as 1 and 2, respectively. Data are expressed as mean ± SEM. Compared to CTL, OUD and MUD had an elevated power of delta/δ (**A**) and theta/θ (**B**) waves but a reduced alpha/α wave (**C**). In contrast, all three waves were reduced in AUD. No change was observed in the beta/β (**D**) or gamma/γ powers (**E**).

easily interpretable in the left-right (Approach 3) and anterior-posterior axis (Approach 4). The concept of electrical axis, which is widely used for EKG [for instance, [31]], was borrowed here for the first time to use in the EEG field. Findings that the left and right spectral powers were normally at an equal level relative to the x-axis could be interpreted as similar EEG activity in the two hemispheres. However, the anterior-posterior axis was no longer in a parallel to x-axis. The observation can be interpreted as that functional impairments were different at two distinct areas. The anterior areas are predominated with neurons for cognition, motivation, and execution while the posteriors are organized with sensory and somatosensory components. Importantly, the anterior-posterior slope could provide a direct comparison of relative changes in axis.

One argument might be that some of information could be eliminated due to the grouped analysis of adjacent electrodes or all 19 electrodes together in Approach 5. Indeed, a conclusion could be partially biased when a single approach is used for data analysis. Thus, we suggest that all five approaches should be included in the data analysis. For instance, EEG signals in patients with AUD became desynchronized but were not statistically significantly different from the CTL when the data were viewed from individual electrodes (Approach 1). The difference became apparent in Approach 2 despite not being significant. Further increases in sample sizes in Approach 3 and 4 resulted in statistically significant differences from the CTL group. This was supported by spectral power analysis across the total cortex with Approach 5, showing that, as sample sizes increased, there were more bands significantly different from the CTL. In this regard, it appears that five approaches of analyses reveal varying information about the data. We suggest that all five approaches should be conducted prior to reaching a conclusion.

We found that powers of each EEG spectrum (i.e., delta/δ, theta/θ, alpha/α, beta/β, or gamma/γ) could be topographically ranked in an order on cortical subregions. A "topomap" has been widely adopted and used for understanding functional connections across cortical networks [32,33]. However, what a normal topomap looks like in a healthy brain is not fully revealed but has been nearly established in recent years. The consensus for alpha/α waves is that they show highest activity (i.e., "hotspot") at occipital subregions [34–37]. A possible explanation for such consistency with alpha/α is that its power is relatively 5–10 times higher than the other spectra and can be reliably observed and identified. The prefrontal or frontal cortices are predominately delta/δ waves [37–39]. Interestingly, hotspots for theta/θ and beta/β powers were located mainly at posterior areas, specifically occipitals, generally in line with previous reports [9,39,40].

Functional connection across cortices is often topographically displayed into gradients [for instance, [39]]. Furthermore, two hemispheres are usually integrated as a single entity. As a matter of fact, EEG signals at electrodes reflect the local dendritic spikes that can be propagated 0.5 mm distance [41] from the scalp [29]. Given that there exists a longitudinal fissure in the skull, it is unlikely that EEG signals at one hemisphere have a spread electrically to the counterpart in long-range spatial manner. This view, however, does not contradict the functional role of the corpus collosum that physically connects two hemispheres. At this point, EEG signals on two hemispheres should be viewed separately and compared whether substances could have a selective effect on one side of the hemispheres [18].

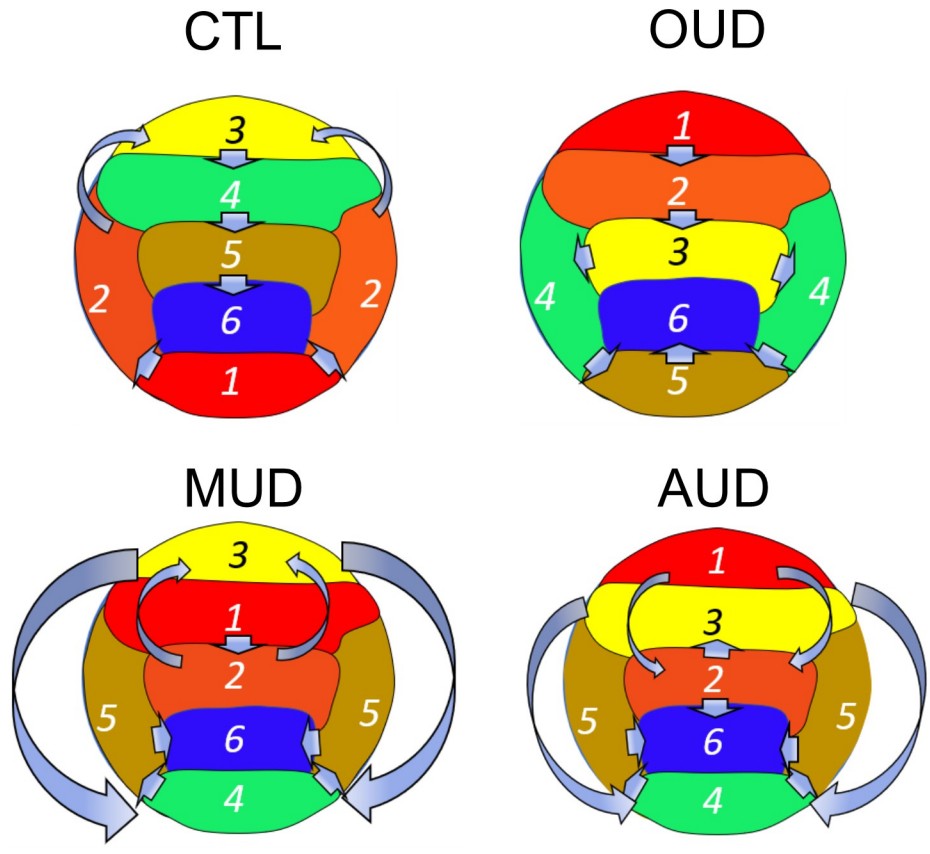

**Fig 7. A**, Gamma/γ power in the cortical subregions altered in drug use disorders. *P<0.05, **P<0.01, and ***P<0.001 vs. CTL, a post-hoc Fisher's PLSD test followed by ANOVA. **B**, Topographic analysis of gamma/γ power. Compared to the CTL, the lowest gamma/γ power still remained at the parietal subregion while the highest power was drifted toward the prefrontal (OUD and AUD) or frontal subregion (MUD).

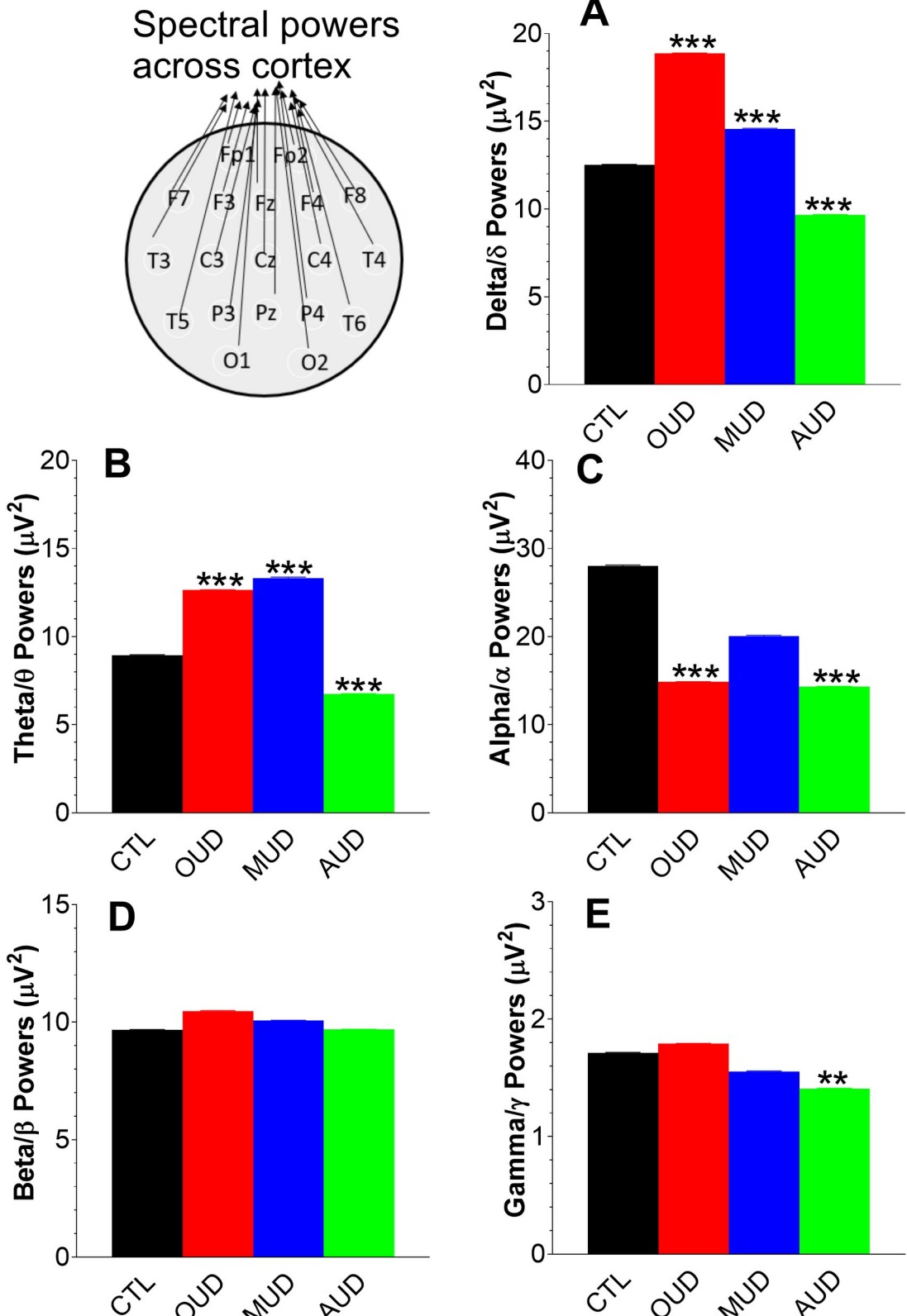

**Fig 8. Analysis of spectral powers as a whole across cortex.** Data are expressed as mean ± SEM. Compared to CTL, spectral powers in patients with SUD (OUD, MUD or AUD) were significantly altered in delta/δ (**A**) and theta/θ (**B**), partly alpha/α (**C**) or gamma/γ (**E**). No effect was observed in the beta/β (**D**). **P<0.01, and ***P<0.001 *vs*. CTL, unpaired t-test.

An interesting finding was that the highest power or hotspot was from the prefrontal area for the delta/δ wave with a characteristic ranking order: prefrontal →frontal →central →parietal →occipital →temporal subregions. In contrast, the theta/θ, alpha/α, beta/β, or gamma/γ wave was found at the occipital subregion with unique rank orders for each spectrum. Taken together, hotspots and/or rank orders of spectral powers could be a physiological feature, which is likely explored as EEG biomarkers to distinguish the healthy people from those with SUD, as discussed, further below.

Delta/δ (1–4 Hz) was the band most vulnerable to be alteration in patients with SUD. As the sample sizes increased, theta/θ (4–8 Hz) waves followed by alpha/α (8–12 Hz) or gamma/γ (25–50 Hz) could be significantly affected. We found that beta/β was the band least sensitive to any effect of substance use disorders, partly in line with previous reports [19,21]. Since etiology of those bands are unknown, it is impossible for us at the present time to interpret why the effects of SUD impacted primarily at the delta/δ wave and secondarily on theta/θ, or what could be the mechanism underlying the beta/β resistance.

A drawback in the present study was that there was too much workload on EEG signal resorting, feature extraction, analysis design and redesign, which were time consuming. It appears these data analyses could be automatically processed with software. Recently, it has been suggested that artificial intelligence (AI) and automatic analysis could apply for some features of EEG signals [for instance, [42]]. To develop such software, the present studies for providing an AI roadmap are two-fold. First, we suggest that AI should analyze EEG signals from at least five aspects, such as individual electrodes, cortical subregions, left-right hemispheres, anterior-posterior axis, and the whole cortex. Second, we suggest that AI should analyze not only EEG amplitude but also other biomarkers, specifically ranking orders of amplitudes and electrical axis. It is no doubt that EEG amplitudes were indicative of mental health alteration by the use of substances. Despite such importance, it cannot exclude the possibility of bias, so other biomarkers and binary classification are needed to be included in the future study by which the conclusion can be alternatively corroborated. Results of the present study demonstrate that spectral powers in the closed-eye state were characteristically altered in not only amplitudes, but also ranking orders and electrical axis in patients with SUD, providing that multiple biomarkers can be evaluated. With an ~1-5min sampling time, AI-driven EEG could emerge as a powerful tool in the future for quick and inexpensive diagnosis on mental health of patients with SUD.

## Supporting information

**S1 File. Raw EEG data recorded from 20 fully anonymized healthy subjects as control (CTL).**
(XLS)

**S2 File. Raw EEG data recorded from 20 fully anonymized patients with opioid use disorder (OUD).**
(XLS)

**S3 File. Raw EEG data recorded from 15 fully anonymized patients with METH use disorder (MUD).**
(XLS)

**S4 File. Raw EEG data recorded from 23 fully anonymized patients with alcohol use disorder (AUD).**
(XLS)

**S5 File. Medical profiles of fully anonymized patients used in the present studies.**
(DOCX)

**S6 File. Effects on theta/θ powers at 19 individual electrodes of patients with OUD, MUD or AUD.** Data were expressed as % CTL **A**, Frontal. **B**, Central. **C**, Temporal. **D**, Parietal. **E**, Occipital. Overall, MUD or OUD theta/θ powers >CTL >AUD. However, OUD, MUD or AUD was not different from the CTL (P>0.05).
(TIF)

**S7 File. Effects on alpha/α powers at 19 individual electrodes of patients with OUD, MUD or AUD.** Data were expressed as % CTL. **A**, Frontal. **B**, Central. **C**, Temporal. **D**, Parietal. **E**, Occipital. Overall, CTL alpha/α power >OUD >MUD >AUD. OUD, MUD or AUD was not different from the CTL (P>0.05).
(TIF)

**S8 File. Effects on beta/β powers at 19 individual electrodes of patients with OUD, MUD or AUD.** Data were expressed as % CTL. **A**, Frontal. **B**, Central. **C**, Temporal. **D**, Parietal. **E**, Occipital. OUD, MUD or AUD was not different from the CTL (P>0.05).
(TIF)

**S9 File. Effects on gamma/γ powers at 19 individual electrodes of patients with OUD, MUD or AUD.** Data were expressed as % CTL. **A**, Frontal gamma/γ powers. **B**, Central gamma/γ powers. **C**, Temporal gamma/γ powers. **D**, Parietal gamma/γ powers. **E**, Occipital gamma/γ powers. OUD, MUD or AUD was not different from the CTL (P>0.05).
(TIF)

## Acknowledgments

The authors would like to thank FHE Health for granting permission to utilize their data for this study. We would like to extend our gratitude to the dedicated staff at the NeuroRehabilitation department at FHE for collecting the data and helping us to make our work as smooth as possible. Additionally, we thank Dr. Ximena Levy for her knowledge over human research review and Jeffrey Clark for his excellent IT support, both of whom are staff at Florida Atlantic University.

## Author Contributions

**Conceptualization:** John J. Callanan, Rui Tao.

**Data curation:** Ibrahim M. Shokry, William To.

**Formal analysis:** Christopher Minnerly, William To.

**Funding acquisition:** Ibrahim M. Shokry, John J. Callanan, Rui Tao.

**Investigation:** Ibrahim M. Shokry, John J. Callanan.

**Methodology:** John J. Callanan.

**Resources:** Christopher Minnerly.

**Supervision:** Rui Tao.

**Validation:** William To.

**Visualization:** Rui Tao.

**Writing – original draft:** Christopher Minnerly, Rui Tao.

**Writing – review & editing:** Ibrahim M. Shokry, William To, John J. Callanan, Rui Tao.

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
