## [Decision Letter · Decision Letter 0]

26 Apr 2021

PONE-D-21-06684

Characteristic changes in EEG spectral powers of patients with opioid-use disorder as compared with those with methamphetamine- and alcohol-use disorders

PLOS ONE

Dear Dr. Rui Tao,

Thank you for submitting your manuscript to PLOS ONE. After careful consideration, we feel that it has merit but does not fully meet PLOS ONE’s publication criteria as it currently stands. Therefore, we invite you to submit a revised version of the manuscript that addresses the points raised during the review process.

We look forward to receiving your revised manuscript.

Kind regards,

Wajid Mumtaz

Academic Editor

PLOS ONE

Journal Requirements:

2. In the Methods section and the online submission form, please provide additional information about the patient records used in your retrospective study.

Specifically, please ensure that you have discussed whether all data were fully anonymized before you accessed them and/or whether the IRB or ethics committee waived the requirement for informed consent.

If patients provided informed written consent to have data from their medical records used in research, please include this information.

3. Thank you for stating the following financial disclosure: 'No'

4. Your ethics statement should only appear in the Methods section of your manuscript.

If your ethics statement is written in any section besides the Methods, please delete it from any other section.

Reviewers' comments:

Reviewer's Responses to Questions

**Comments to the Author**

1. Is the manuscript technically sound, and do the data support the conclusions?

Reviewer #1: Yes

Reviewer #2: Partly

2. Has the statistical analysis been performed appropriately and rigorously? 

Reviewer #1: Yes

Reviewer #2: No

3. Have the authors made all data underlying the findings in their manuscript fully available?

Reviewer #1: Yes

Reviewer #2: No

4. Is the manuscript presented in an intelligible fashion and written in standard English?

Reviewer #1: Yes

Reviewer #2: Yes

5. Review Comments to the Author

Reviewer #1: This is a well designed and intriguing study.

I think the results would be even more relevant and valuable if there was an analysis done of the EEG spectra in wakefulness vs REM sleep vs NREM sleep. At least there should be a comparison between wakefulness and sleep as alcohol, opiates and stimulants can affect neurophysiological function in sleep differently than they do in wakefulness

Reviewer #2: The topic of the paper is interesting. However, the statistical analysis should be improved. Comments are listed below.

1. Data availability: Data location has not been listed. As such, data is not available.

2. Data collection/preprocessing: Only 10-minutes of resting state EEG sampled at 256 Hz is analyzed. Were the data collected only once? Did the subjects return for multiple recordings? Were their tasks performed? A single recording at resting-state is better than no recording, but the conclusions may be limited by one-time recording due to non-stationary behavior of the brain signals. Were the powerline noise removed from the data?

3. Selection of Subjects: Out of 350 Opioid, 450 Alcohol and 100 Meth, selection of only about 20 subjects per group seems to be limited by the control group. This is the main weakness of the paper. The authors can divide the A, M and O groups to smaller groups of 20 subjects and see if an average of all subgroups match with the current conclusions of the paper. This may increase statistical significance of the results.

4. While absolute band powers have been analyzed, it may be useful to analyze relative band powers. These have played a role in discriminating two classes of EEG signals such as pre-ictal and inter-ictal in epilepsy. Relative band powers can be calcluated by dividing every subjects band power by the total power spectral density. See for example,

Z. Zhang et al, "Low-Complexity Seizure Prediction From iEEG/sEEG using Spectral Power and Ratios of Spectral Power," 2016

5. While the statistical behavior is useful and should be published, the community has focused on binary classification. This could be considered as future work. As one example of such classification, the authors may refer to:

B. Sen et al., "Sub-Graph Entropy based Network Approaches for Classifying Adolescent Obsessive-Compulsive Disorder from Resting-State Functional MRI," 2020

6. PLOS authors have the option to publish the peer review history of their article (what does this mean?). If published, this will include your full peer review and any attached files.

Reviewer #1: **Yes: **Hrayr Attarian

Reviewer #2: No

---

## [Author Response · Author response to Decision Letter 0]

14 Jun 2021

Response to Dr. Hrayr Attarian

Q5. ……. the results would be even more relevant and valuable if there was an analysis done of the EEG spectra in wakefulness vs REM sleep vs NREM sleep. At least there should be a comparison between wakefulness and sleep as alcohol, opiates and stimulants can affect neurophysiological function in sleep differently than they do in wakefulness

Reply: We totally agree with the importance to show the difference between wakefulness and sleep in patients with opioid use disorder in contrast to those with alcohol or stimulant use disorders. However, there is no sleep data available for us. Therefore, we could not conduct the comparative study. The comments are very valuable for future direction. 

The retrospective EEG data used in the present studies refers only at the eye-closed (wakeful) state. 

Response to Reviewer #2: 

Q6. Data availability: Data location has not been listed. As such, data is not available.

Reply: Thanks for the comment. In this revision, every raw data was attached to supplementary sections S1-S5. 

Q7. ……. Were the data collected only once? Did the subjects return for multiple recordings? 

Reply: Only one time of data prior to rehab treatment was approved to be used in the present study. Patients did return after rehab treatment. However, data after rehab treatment were not approved for disclosure.

Q8. Were their tasks performed? 

Reply: No tasks performed. The eye closed, and subject seated calmly without any activity or movement.

Q9……Were the powerline noise removed from the data?

Reply: Yes, it was removed. Additionally, artifacts (non-EEG) were also removed from the data, as described on pg6 line 149-151 “…….. raw data was edited using the editing tool within the NeuroGuide software to remove physical artifacts (including eye movement, jaw movement, and gross movement)……”

Q10. Out of 350 Opioid, 450 Alcohol and 100 Meth, selection of only about 20 subjects per group seems to be limited by the control group. This is the main weakness of the paper. 

Reply: We appreciate the reviewer’s concern. However, most patients in our database were found to be multi-drug users. Therefore, most of them should be excluded from the study. Patients with neurodegenerative diseases (e.g., PD and MD) should also be excluded. Only a few dozen patients met the criteria for meaningful retrospective data analysis. 

Our primary goal in this study was to characterize EEG in patients with opioid use disorder (OUD). For this reason, 20 OUD patients were compared with sex- and age-matched healthy control. We believe that readers might feel more interested in knowing the difference in EEG between OUD and stimulants (MUD) or alcohol (AUD). Therefore, we compared OUD with MUD or AUD. Although we could not get sex- and age-matched (stimulant/MUD or alcohol/AUD) data, we think the results were still valuable, even though, we had only 15 MUD patients.

Q11. The authors can divide the A, M and O groups to smaller groups of 20 subjects and see if an average of all subgroups match with the current conclusions of the paper. This may increase statistical significance of the results.

Reply: We conducted a statistical analysis with a smaller number of cases as suggested to see if statistical significance of the results would increase or not. Because we had only 15 MUD cases, we had to remove 5 cases out of the CTL (C4, C7, C14, C12, and C15), 5 from the opioids (O7, O15, O17, O18, and O34), and 7 out of the alcohol (A3, A8, A19, A25, A27, and A28) to have 15 cases per every group. All groups were analyzed. Please see the table below.

 CTL (N=15) OUD (N=15) MUD (N=15) AUD (N=15) 

Age (years) 29 (�8) 30 (�9) 29 (�8) 33 (�6) 

Sex (M/F) 8/7 8/7 11/4 10/5 

Duration of substances used (years) 0 6 (�4) 5 (�3) 6 (�7) 

We started to re-examine delta/� powers presented in fig 3B. The ANOVA was used to analyze 4 groups of absolute delta/� values of the F3 electrode, revealing significant difference between 4 groups, showing F(3,56) =3.203, P =0.03. We also analyzed the percentage values and had the same results. We next conducted post-hoc test analysis revealing that changes in opioids/OUD, stimulants/MUD or alcohol/OUD were not different from CTL (see figure below). The statistical significance of the results decreased. Given that increases in numbers of cases would improve the significance of statistical analysis, we decided to keep the original analysis (i.e., 20 cases in CTL; 20/opioid, 15/stimulants, and 23/alcohol) showing F(3,74) =6.07, P). =0.0009.

To clarify the question for readers, we added three new sentences on pg11 line251-254, stating that ‘The normalized data appear to have the same response pattern as……absolute value’

In addition to F3, all 19 electrodes were re-examined using smaller numbers as shown below. Similarly, figures C’-G’ displayed the same tendency as figure 3C-G in the manuscript although case numbers reduced. However, no electrode was found to have statistical significance. For this reason, we kept the case numbers in the manuscript as was.

Q12. While absolute band powers have been analyzed, it may be useful to analyze relative band powers. These have played a role in discriminating two classes of EEG signals such as pre-ictal and inter-ictal in epilepsy. Relative band powers can be calculated by dividing every subjects band power by the total power spectral density. See for example,

Z. Zhang et al, "Low-Complexity Seizure Prediction From iEEG/sEEG using Spectral Power and Ratios of Spectral Power," 2016.

Reply: This is also a good point raised by the reviewer. Analysis of absolute values could be simple but potentially create data bias due to individual differences, which was correctly emphasized by Zhang et al 2016. To do this, EEG changes in seizure attack could be normalized to the pre-seizure EEG powers, which is a perfect approach for the seizure analysis.

Regarding our patients with substance use disorders (SUD), pre-SUD data were not available in the database. Therefore, we could not calculate changes relative to pre-SUD levels when they were still mentally healthy. Fortunately, there were 20 healthy CTL included in the study. Those CTL data could be used as pre-SUD data. Specifically, in our manuscript, relative changes (% CTL) were determined with CTL EEG powers, displayed in fig 3A-G, and also supplementary S2-5. In other words, our data analysis in the manuscript considered both absolute and relative values. We found that statistical analyses over relative changes were the same as the absolute values, indicating no difference. Absolute values were used in figures 4-8 for a direct comparison of our results with other labs (see details on pg11 line251-254)

Alternatively, relative powers of individual/group electrodes can be divided by the total powers of all 19 electrodes. Please note that the total powers presented in fig 8, showed significant difference in EEG between users. Although we haven’t divided individual/group electrodes by the total powers, we will include this idea in future analysis.

Q13. While the statistical behavior is useful and should be published, the community has focused on binary classification. This could be considered as future work. As one example of such classification, the authors may refer to:

B. Sen et al., "Sub-Graph Entropy based Network Approaches for Classifying Adolescent Obsessive-Compulsive Disorder from Resting-State Functional MRI," 2020.

Reply: We appreciate this suggestion.

---

## [Decision Letter · Decision Letter 1]

16 Jul 2021

PONE-D-21-06684R1

Characteristic changes in EEG spectral powers of patients with opioid-use disorder as compared with those with methamphetamine- and alcohol-use disorders

PLOS ONE

Dear Dr. Rui Tao,

Thank you for submitting your manuscript to PLOS ONE. After careful consideration, we feel that it has merit but does not fully meet PLOS ONE’s publication criteria as it currently stands. Therefore, we invite you to submit a revised version of the manuscript that addresses the points raised during the review process.

We look forward to receiving your revised manuscript.

Kind regards,

Wajid Mumtaz

Academic Editor

PLOS ONE

Journal Requirements:

Reviewers' comments:

Reviewer's Responses to Questions

**Comments to the Author**

1. If the authors have adequately addressed your comments raised in a previous round of review and you feel that this manuscript is now acceptable for publication, you may indicate that here to bypass the “Comments to the Author” section, enter your conflict of interest statement in the “Confidential to Editor” section, and submit your "Accept" recommendation.

Reviewer #1: All comments have been addressed

Reviewer #2: (No Response)

2. Is the manuscript technically sound, and do the data support the conclusions?

Reviewer #1: Yes

Reviewer #2: Yes

3. Has the statistical analysis been performed appropriately and rigorously? 

Reviewer #1: Yes

Reviewer #2: N/A

4. Have the authors made all data underlying the findings in their manuscript fully available?

Reviewer #1: Yes

Reviewer #2: Yes

5. Is the manuscript presented in an intelligible fashion and written in standard English?

Reviewer #1: Yes

Reviewer #2: Yes

6. Review Comments to the Author

Reviewer #1: Thank you for thoroughly addressing all the reviewer concerns. I have no further comments about this paper

Reviewer #2: Statements from the response should be included in main part of the paper. future directions with respect to features and classifiers should be pointed out.

7. PLOS authors have the option to publish the peer review history of their article (what does this mean?). If published, this will include your full peer review and any attached files.

Reviewer #1: **Yes: **Hrayr Attarian

Reviewer #2: No

---

## [Author Response · Author response to Decision Letter 1]

11 Aug 2021

Response to editorial requirement

Q1. Please review your reference list to ensure that it is complete and correct. If you have cited papers that have been retracted, please include the rationale for doing so in the manuscript text, or remove these references and replace them with relevant current references. Any changes to the reference list should be mentioned in the rebuttal letter that accompanies your revised manuscript. If you need to cite a retracted article, indicate the article’s retracted status in the References list and also include a citation and full reference for the retraction notice.

Reply: The references cited in the manuscript has been reviewed in a one to one manner. Specifically, the references were copy-pasted to the NIH PUBMED website. There was no information indicating retraction of references cited in the manuscript.

Response to reviewer #2

Q2. …the response should be included in main part of the paper. Future study direction with respect features and classifiers should be pointed out

Reply: the response has been included into the context of the manuscript. Details are as follows.

Questions and concerns Pages, lines and response located in the manuscript

Were the data collected only once? Did the subjects return for multiple recordings?

 on pg6 lines 150-151 

“Only one time of data prior to rehab treatment was approved by the IRAs to be used in the present study”

“Were their tasks performed?” on pg6 lines 135-136 

“Eyes closed, and subject seated calmly without any activity or movement”

“Were the powerline noise removed from the data?

 on pg6 line 152-154

“…….. raw data was edited using the editing tool within the NeuroGuide software to remove physical artifacts (including eye movement, jaw movement, and gross movement)…

The authors can divide the A, M and O groups to smaller groups of 20 subjects and see if an average of all subgroups match with the current conclusions of the paper. This may increase statistical significance of the results On pg15-16

……(see details in fig 3, and also supplementary S1-S4). One explanation for such failure could be that the retrospective analysis was not designed as a one-to-one matched case control study. It was likely to be that the statistical power would be increased if the subgroups were matched. To test such possibility, subjects were reduced to 15 cases/subgroup for the age- and sex-matched design. Delta/� powers presented in fig 3B were re-examined with the ANOVA, revealing significant difference between 4 groups, F(3,56) =3.203, P =0.03. However, the post-hoc analysis revealed that changes in OUD, MUD or AUD were not different from CTL, suggesting the one-to-one matched design could not fully explain the failure at statistical analysis.

Alternatively, a single electrode covered only small brain areas in terms of EEG spatial sizes and thus relatively small numbers of neurons affected….

While absolute band powers have been analyzed, it may be useful to analyze relative band powers. These have played a role in discriminating two classes of EEG signals such as pre-ictal and inter-ictal in epilepsy. Relative band powers can be calculated by dividing every subjects band power by the total power spectral density. See for example,

Z. Zhang et al, "Low-Complexity Seizure Prediction From iEEG/sEEG using Spectral Power and Ratios of Spectral Power," 2016. On pg 10 line247-pg11 line 253

Next, the EEG waves were transformed into amplitude powers expressed as µV2, as shown in the left panel of Fig 3B. The difference in delta/� amplitude powers was statistically significant [F(3, 74) =6.07, P =0.0009]. However, post-hoc analysis indicates that only OUD, but not MUD or AUD, reached statistical significance difference from the CTL. Analysis of absolute values could be simple but potentially create data bias due to individual differences. To minimize such possibility, data were normalized into %CTL.

While the statistical behavior is useful and should be published, the community has focused on binary classification. This could be considered as future work. As one example of such classification, the authors may refer to:

B. Sen et al., "Sub-Graph Entropy based Network Approaches for Classifying Adolescent Obsessive-Compulsive Disorder from Resting-State Functional MRI," 2020. On pg19, line 496-498

….Despite such importance, it cannot exclude the possibility of bias, so other biomarkers and binary classification are needed to be included in the future study by which the conclusion can be alternatively corroborated…

---

## [Decision Letter · Decision Letter 2]

26 Aug 2021

Characteristic changes in EEG spectral powers of patients with opioid-use disorder as compared with those with methamphetamine- and alcohol-use disorders

PONE-D-21-06684R2

Dear Dr. Rui Tao,

We’re pleased to inform you that your manuscript has been judged scientifically suitable for publication and will be formally accepted for publication once it meets all outstanding technical requirements.

Kind regards,

Wajid Mumtaz

Academic Editor

PLOS ONE

Additional Editor Comments (optional):

Reviewers' comments:

Reviewer's Responses to Questions

**Comments to the Author**

1. If the authors have adequately addressed your comments raised in a previous round of review and you feel that this manuscript is now acceptable for publication, you may indicate that here to bypass the “Comments to the Author” section, enter your conflict of interest statement in the “Confidential to Editor” section, and submit your "Accept" recommendation.

Reviewer #2: All comments have been addressed

2. Is the manuscript technically sound, and do the data support the conclusions?

Reviewer #2: Yes

3. Has the statistical analysis been performed appropriately and rigorously? 

Reviewer #2: Yes

4. Have the authors made all data underlying the findings in their manuscript fully available?

Reviewer #2: Yes

5. Is the manuscript presented in an intelligible fashion and written in standard English?

Reviewer #2: Yes

6. Review Comments to the Author

Reviewer #2: Paper is acceptable. I would still suggest to cite the references suggested in the initial review as directions for future work.

7. PLOS authors have the option to publish the peer review history of their article (what does this mean?). If published, this will include your full peer review and any attached files.

Reviewer #2: No

---

## [Editor Report · Acceptance letter]

31 Aug 2021

PONE-D-21-06684R2 

Characteristic changes in EEG spectral powers of patients with opioid-use disorder as compared with those with methamphetamine- and alcohol-use disorders 

Dear Dr. Tao:

I'm pleased to inform you that your manuscript has been deemed suitable for publication in PLOS ONE. Congratulations! Your manuscript is now with our production department. 

Kind regards, 

on behalf of

Dr. Wajid Mumtaz 

Academic Editor

PLOS ONE